# A Conserved Cysteine Residue in Coxsackievirus B3 Protein 3A with Implication for Elevated Virulence

**DOI:** 10.3390/v14040769

**Published:** 2022-04-07

**Authors:** Martin Voss, Sandra Pinkert, Meike Kespohl, Niclas Gimber, Karin Klingel, Jan Schmoranzer, Michael Laue, Matthias Gaida, Peter-Michael Kloetzel, Antje Beling

**Affiliations:** 1Institute of Biochemistry, Charité–Universitätsmedizin Berlin, Corporate Member of Freie Universität Berlin and Humboldt-Universität zu Berlin, 10117 Berlin, Germany; martin.voss81@gmx.net (M.V.); sandra.pinkert@charite.de (S.P.); meike.kespohl@charite.de (M.K.); niclas.gimber@charite.de (N.G.); jan.schmoranzer@charite.de (J.S.); p-m.kloetzel@charite.de (P.-M.K.); 2Deutsches Zentrum für Herz-Kreislauf-Forschung, Partner Side Berlin, 10117 Berlin, Germany; 3Charité–Universitätsmedizin Berlin, Corporate Member of Freie Universität Berlin and Humboldt-Universität zu Berlin, Advanced Medical Bioimaging Core Facility, 10117 Berlin, Germany; 4Cardiopathology, Institute for Pathology and Neuropathology, University of Tübingen, 72016 Tübingen, Germany; karin.klingel@med.uni-tuebingen.de; 5Robert Koch Institute, Advanced Light and Electron Microscopy (ZBS 4), 13353 Berlin, Germany; lauem@rki.de; 6Institute of Pathology, University Medical Center Mainz, JGU-Mainz, 55131 Mainz, Germany; matthias.gaida@unimedizin-mainz.de; 7Research Center for Immunotherapy, University Medical Center Mainz, JGU-Mainz, 55131 Mainz, Germany; 8Joint Unit Immunopathology, Institute of Pathology, University Medical Center, JGU-Mainz and TRON, Translational Oncology at the University Medical Center, 55131 Mainz, Germany

**Keywords:** enterovirus, coxsackievirusB3, infection, animal model, virulence regulation

## Abstract

Enteroviruses (EV) are implicated in an extensive range of clinical manifestations, such as pancreatic failure, cardiovascular disease, hepatitis, and meningoencephalitis. We recently reported on the biochemical properties of the highly conserved cysteine residue at position 38 (C38) of enteroviral protein 3A and demonstrated a C38-mediated homodimerization of the Coxsackievirus B3 protein 3A (CVB3-3A) that resulted in its profound stabilization. Here, we show that residue C38 of protein 3A supports the replication of CVB3, a clinically relevant member of the enterovirus genus. The infection of HeLa cells with protein 3A cysteine 38 to alanine mutants (C38A) attenuates virus replication, resulting in comparably lower virus particle formation. Consistently, in a mouse infection model, the enhanced virus propagation of CVB3-3A wt in comparison to the CVB3-3A[C38A] mutant was confirmed and found to promote severe liver tissue damage. In contrast, infection with the CVB3-3A[C38A] mutant mitigated hepatic tissue injury and ameliorated the signs of systemic inflammatory responses, such as hypoglycemia and hypothermia. Based on these data and our previous report on the C38-mediated stabilization of the CVB3-3A protein, we conclude that the highly conserved amino acid C38 in protein 3A enhances the virulence of CVB3.

## 1. Introduction

Enteroviruses (EV) comprise a large genus of single-stranded RNA viruses that cause a wide spectrum of diseases with potentially lethal progression. During periods of high EV prevalence, newborn infants and young children are at particular risk of developing life-threatening septic syndromes [1,2]. In addition to their significant pathology in humans, EV are also endemic among domestic pigs and occasional outbreaks of severe porcine EV infection are reported worldwide. For some human EV strains, there are also reports showing transmission to pigs and vice versa [3]. Among the various clinically relevant EV, the human EV coxsackievirus B3 (CVB3) is a well-studied pathogen and the clinical manifestation of CVB3 infection can extend to severe conditions such as myocarditis, pancreatitis, meningoencephalitis, and hepatitis, with potentially fatal outcomes [2]. The clinical picture in humans can be mimicked in mice, and here the severity of CVB3 infection, under the influence of the host’s innate antiviral effector responses, is associated with extensive virus-mediated cytotoxicity—e.g., in the pancreas [4], liver [5,6], and heart tissue [7,8]—and is accompanied by systemic inflammation.

Both cell culture and mouse infection models have revealed mechanisms that evolved to support the viral replication process and defined immune escape mechanisms executed by the virus to expedite its growth. Similar to other RNA viruses, CVB3 manipulates cellular proteins and lipids for the formation of replication organelles, a process that optimizes the concentration and localization of the viral genome and viral and cellular proteins and supports the assembly and release of viral particles from infected cells [9]. For CVB3, the respective membrane remodeling in infected cells and the generation of replication organelles requires the non-structural viral protein 3A. Protein 3A is formed by the cleavage of the viral precursor protein 3AB and consists of a soluble N-terminus and a hydrophobic C-terminus, the latter forming a single transmembrane helix with membranes of the endoplasmic reticulum (ER) and Golgi apparatus. Protein 3A not only promotes the replication process by the formation of replication organelles but also mediates immune escape functions in CVB3 infection. As a transmembrane protein, 3A can block the anterograde traffic of host proteins from the ER to the Golgi complex, contributing to the reduced presentation of MHC molecules at the cellular surface and limiting the secretion of anti-viral cytokines [10,11,12,13]. Mutations in the 3A coding region give rise to viruses defective in viral RNA synthesis and incapable of blocking ER-to-Golgi traffic, but a specific function of the 3A protein in the viral replication cycle has not yet been demonstrated [14,15].

In infected cells, the N-terminal domain of protein 3A resides on the cytosolic side of cellular membranes, where it recruits host proteins such as guanine nucleotide exchange factor GBF1 as well as acyl-CoA-binding domain-containing protein 3 (ACBD3). The N-terminus of protein 3A can interact with the ACBD3 GOLD domain, promoting heterodimer formation, which has been shown to promote viral replication [16,17]. Moreover, there are also protein-protein interactions between individual 3A proteins and protein X-ray crystallography previously defined residues L25, V34, and Y37 to form the hydrophobic core of the protein 3A dimerization interface [16]. At the side of viral replication, ACBD3-protein 3A heterodimers/heterotetramers recruit the lipid kinase phosphatidylinositol 4-kinase-β (PI4KB), a Golgi-localized lipid kinase that phosphorylates phosphatidylinositol to yield phosphatidylinositol-4-phosphate (PI4P) lipids [18,19]. The 3A-ACBD3-PI4KB route represents a major mechanism of PI4KB recruitment to the sites of EV replication [19]. PI4P-modified lipids attract oxysterol-binding protein (OSBP) and thereby form membrane contact sites between the ER and replication organelles, triggering an accumulation of cholesterol that is necessary for efficient viral genome replication [9].

In addition to L25, V34, and Y37, other residues of protein 3A can mediate homodimer formation as well, and this may influence the replication process by as-yet-unknown mechanisms. As an example, protein 3A from poliovirus, another member of the EV genus, forms a symmetric homodimer via the soluble N-terminal domain [20]. We recently demonstrated that CVB3 protein 3A forms SDS-resistant homodimers [21] via a DTT-sensitive disulfide bridge between cysteine residues at position 38 (C38), thereby increasing its stability [22]. The conservation of 3A-C38 among various representatives of the EV genus suggests that the functional properties of this cysteine residue are a common feature of EV protein 3A [22]. These aspects prompted us to investigate whether cysteine 38 of CVB3 protein 3A plays a role in infection, and our experiments documented a pro-viral function of 3A-C38 that occurs by it supporting the production of viral particles and enhancing cytotoxicity.

## 2. Results

### 2.1. Cells Infected with Protein 3A C38A Mutant and Wild-Type Virus Display Similar Ultrastructural Changes

The structural modelling of C38-linked protein 3A homodimers implies an efficient recruitment of ACBD3 by protein 3A irrespective of this disulfide bridge [22]. To investigate whether protein 3A-C38 indeed has, as predicted, no effect on the recruitment of ACBD3 to cellular membranes in CVB3-infected cells, C38 of CVB3 3A was mutated to alanine by site-directed mutagenesis, generating CVB3-3A[C38A]. This mutation rendered protein 3A inert to a C38-mediated formation of a disulfide bridge (Figure 1A). For the assessment of the effect of protein 3A-C38 on ACBD3 distribution during infection, extracts of cells infected with CVB3 3A wild-type (wt) or CVB3 3A[C38A] virus were subjected to differential centrifugation to obtain a medium-speed membrane pellet (16 k), a high-speed membrane pellet (120 k), and a supernatant fraction representing the cytosolic fraction. With this approach, we could document the cellular distribution of viral and host proteins in the different compartments during infection. We previously demonstrated that the CVB3 proteins are most abundantly localized in the 16k membrane pellet [21], indicating that this fraction contains replication organelles. The redistribution of ACBD3 in CVB3-infected cells from the cytosolic fraction to the 16k membrane fraction, which contains protein 3A, was similar for both the wt 3A and the mutant C38A variants (Figure 1B). Thus, it is most likely that wt and mutated protein 3A can efficiently interact with ACBD3 proteins, recruiting them independently of 3A-C38 to cellular membranes and causing similar PI4KB-dependent alterations in the lipid composition during CVB3 infection. We also examined whether the different effects of the wt and mutant 3A with regard to the disulfide bridge formation of protein 3A could affect the generation of the membranous vesicles on which virus RNA replication occurs. To investigate the effect of the 3A C38A mutant on the rearrangement of cellular membranes during viral infection, the ultrastructure of cells infected with wt and CVB3-3A[C38A] mutant was examined using electron microscopy. This experiment documented similar ultrastructural changes with a cluster of vesicles in the cytoplasm, where Golgi stacks are found in uninfected cells (Figure 1C–H), indicating that the 3A[C38A] mutation did not inhibit the membrane rearrangements induced by CVB3 infection. The formation of replication organelles for cells infected by the mutant strain was similar to that found in cells infected with CVB3-3A wt. This finding confirms that 3A-C38 is not essential for the initiation phase that leads to tubulovesicular membrane structures and the formation of replication organelles in CVB3-infected cells.

### 2.2. Cysteine 38 of Protein 3A Affects Its Subcellular Distribution in Infected Cells

The close proximity that we documented in CVB3-infected cells for the Golgi apparatus, emerging replication organelles, and viral particles (Figure 1E–H) concurs with the capacity of protein 3A to interfere with cellular protein secretion via the inhibition of the ER to Golgi transport, hence promoting viral replication [13]. We recently demonstrated that 3A-C38 elevates the intensity of protein 3A signals at cellular membranes, suggesting that the capacity to form C38-linked protein 3A homodimers might increase the availability of this viral protein at replication organelles. To assess the reported effect of C38 on protein 3A in the context of CVB3 infection, we made use of the C38A mutant that was generated by the site-directed mutagenesis of recombinant CVB3 encoding in addition to EGFP in its genome (Figure 1A). An immunofluorescence analysis of CVB3-3A wt-infected HeLa cells showed that 5 h after the start of infection, corresponding to the exponential phase of the replication cycle, protein 3A is highly concentrated in few bright clusters (Figure 2A). The staining of 3A in CVB3-3A[C38A]-infected cells on the other hand is localized, with a lower intensity in less-defined clusters merging into bigger structures, distributed all about the cell. A quantitative comparison of anti-3A staining intensities showed a significant difference in intensity distribution between CVB3-3A wt and CVB3-3A[C38A] (Figure 2B). Infection with CVB3 encoding EGFP results in GFP expression after 5 h. Cells with similar GFP fluorescence and 3A staining intensity, both being indicators of CVB3 infection in their respective HeLa cells, were selected for CVB3-3A wt and CVB3-3A[C38A] (Appendix A). An analysis of anti-3A immunofluorescence using autocorrelation-based image correlation spectroscopy showed bigger areas of staining clusters in the case of protein 3A for CVB3-3A[C38A]-infected cells, thus confirming that cysteine 38 of protein 3A at least partially directs its subcellular localization towards more condensed membrane structures, as shown for the ectopic expression of protein 3A-C38A variants [22]. The cellular alterations seen in CVB3-3A wt-infected cells that we attributed to 3A-C38 suggested a more pronounced availability of protein 3A at the side of viral replication, which is in agreement with the enhanced stability of protein 3A as a C38-linked homodimer, as previously shown.

### 2.3. Cysteine 38 of Protein 3A Supports Virus Replication

The 3A-C38-mediated increased the stability for protein 3A [22], and the detection of a higher protein 3A signal intensity at specific foci in infected cells (Figure 2) prompted us to explore a role of the highly conserved 3A-C38 residue in virus replication. Virus with the C38A mutation was viable. We infected cells with the CVB3 3A-C38A mutant and compared its replication with that of the wt strain in more detail. To ascertain that the obtained effects on viral replication were due to a disulfide bond in CVB3 3A protein and did not constitute an impaired non-covalent interaction mediated by the polar properties of the cysteine residue, we also analyzed a C38S mutation of CVB3 protein 3A (CVB3-3A[C38S]), with serine being chemically related to cysteine. The assessment of viral replication showed that the effects of C38A and C38S are similar, resulting in ~50% lower viral titers compared to CVB3 encoding wt protein 3A (Figure 3A). Five hours after infection at MOI 1, the virus replication of CVB3-3A[C38A] was markedly reduced compared to CVB3-3A wt, as confirmed by fluorescence microscopy and the flow cytometry-based quantification of co-expressed GFP (Figure 3B,C). The difference in replication efficiency between CVB3-3A wt and CVB3-3A[C38A] was also observed when infecting cells at MOI 0.1 for 12 to 36 h, a period that entails several replication cycles (Figure 3D,E). The impaired viral replication of CVB3-3A[C38A] was also reflected by less intense GFP staining (Figure 3D) as well as lower virus titers (Figure 3E). The importance of protein 3A-C38-mediated effects in viral replication is supported by the fact that the levels of the viral proteins 3D and VP1 in the 16k membrane pellet were up to 60% lower during infection with CVB3 encoding 3A[C38A]. The difference seen for monomeric protein 3A and 3AB was even more pronounced, with nearly 75% lower expression levels seen for CVB3-3A[C38A] (Figure 3F).

Consistent with the functional role of the C38 residue in viral replication in CVB3 infection, the cytopathic effect was more pronounced for cells that were infected with CVB3-3A wt in comparison to the CVB3 3A[C38A] mutant (Figure 3G).

### 2.4. Infection with the CVB3-3A[C38A] Mutant Mitigates Viral Pathology

To test whether protein 3A cysteine 38 affects the virulence of the pathogenic CVB3 H3 strain in vivo, C57BL/6 mice were infected with CVB3 encoding wt (CVB3-3A wt) or C38A protein 3A (CVB3-3A[C38A]) and sacrificed after three days. The assessment of the amount of infectious viral particles for this point in time by plaque assay showed significantly higher virus concentrations in the spleen, pancreas, heart, and liver of animals infected with CVB3-3A wt (Figure 4A), thus corroborating the results obtained in cell culture and highlighting the functional importance of 3A cysteine 38. Correspondingly, the progression documented for the loss of body weight was less severe in mice infected with CVB3-3A[C38A], reaching a similar decline to that seen for the CVB3-3A wt strain after three days (Figure 4B). The infection of C57BL/6 mice with CVB3-3A wt triggered a hypothermic stage after three days, and this characteristic of a systemic inflammatory response (SIR) after virus infection was substantially milder in CVB3-3A[C38A]-infected mice (Figure 4C). CVB3-induced SIR was accompanied by hypoglycemia, as evidenced by the low blood glucose concentration and decreased liver glycogen seen in infected mice (Figure 4D–F). In comparison to the CVB3-3A wt strain, infection with CVB3-3A[C38A] profoundly attenuated this catabolic disturbance, resulting in higher serum glucose levels and an enhanced detection of hepatic glycogen. The profiling of virus-triggered liver tissue damage by histology and pathobiochemistry revealed multifocal necrosis and inflammation three days after infection, which was accompanied by enhanced serum activity of liver enzymes such as alkaline phosphatase (AP) (Figure 4G–I). The liver tissue damage was less prominent in mice that were infected with CVB3-3A[C38A], emphasizing the functional impact of 3A cysteine 38 in infection. Altogether, the in vivo infection study demonstrated that CVB3 3A cysteine 38 supports viral replication and presents a virulence trait with disease-exacerbating effects in mice.

## 3. Discussion

Infection by EV causes extensive cellular reorganization, including a protein 3A-mediated generation of replication organelles and the recruitment of cellular proteins such as GBF1, ACBD3, and PI4KB, all supporting viral RNA synthesis and virion assembly [9,13,24]. In this study, we investigated whether the highly conserved C38 of protein 3A, which supports the disulfide-linked homodimer formation of EV protein 3A and increases its stability [22], influences the virulence of CVB3. To explore the effect of cysteine residue 38 in protein 3A on CVB3 replication, we used a virus with a C38A mutation in 3A created by the site-directed mutagenesis of the CVB3 cDNA. The experimental detection of the CVB3 3A-[C38A] mutant demonstrates that the function of 3A-C38 to form homodimers at the cytosolic leaflet of membranes is not an absolute prerequisite for viral replication. In fact, the formation of viral particles under conditions where protein 3A cannot form DTT sensitive disulfide bridges (CVB3 3A-[C38A]) concurs with the efficient hijacking of the host ACBD3 protein to the sites of viral replication as well as the generation of replication organelles, as shown by ultrastructural microscopy in cells infected with CVB3 3A-[C38A]. Differential centrifugation and immunoblotting experiments together with immunofluorescence and electron microscopy demonstrated that protein 3A, expressed in the context of viral infection, co-localizes with membranes derived from the Golgi compartment irrespective of 3A-C38. In cells that are either infected with the wt virus or the 3A-[C38] mutant, vesicles accumulated in the region of the cytoplasm where Golgi stacks are found in uninfected cells, indicating that the ER-to-Golgi traffic at the step of vesicle formation is not affected by 3A-C38. These results confirm the recent demonstration of the intact interaction of the mutant 3A-C38A protein with cellular membranes [22].

The ectopic expression of 3A or infection with CVB3 drastically alters the intracellular localization of ACBD3. ACBD3 serves as a hub for various protein–protein interactions, and hence it participates in a plethora of cellular signaling pathways [25]. When cells were infected with CVB3, ACBD3 was found to co-localize with membrane-localized 3A protein. The structural model that we presented recently for the disulfide-linked protein 3A homodimer [22] demonstrated that its protomer–protomer interface is completely different from the ACBD3–protein 3A interfaces shown in [16], and the respective protomer structures themselves were not altered by the C38-mediated dimerization of the protein 3A. From the perspective of this structural homodimer model, 3A protein homodimer formation associated with a cysteine bridge does not change the complex formation with ACBD3 [22]. Consistently, the recruitment of ACBD3 to the membrane in infected cells was not affected by protein 3A’s ability to form C38-C38-linked homodimers. The intact interaction of both wt 3A and C38A 3A with ACBD3 was also confirmed by the lack of any substantial effect of 3A-C38 on the formation of replication organelles, confirming that C38-C38-linked homodimers are not essential for viral replication.

On the other hand, the loss of the capability of 3A to be stabilized by C38-mediated homodimerization reduced the yield of 3A-[C38A] mutant virus in single- and multiple-replication cycle infections. Mouse infection experiments with wt CVB3 and the mutant CVB3 3A-[C38A] confirmed the less efficient replication of the mutant strain, as indicated by reduced viral titers in different tissues. From a pathophysiological perspective and based on the expression of 3A-C38, the infection of mice with CVB3 resulted in elevated cellular cytotoxicity, promoted severe liver tissue injury, and exacerbated the systemic signs of viral infection and thereby triggered inflammation. Based on these findings, we propose a functional role of the property to form C38-mediated disulfide-linked protein 3A dimers, which can enhance the protein 3A stability, thereby supporting viral replication. Multimeric protein complexes, being resistant to proteasomal degradation, might contribute to an increased local concentration of host and viral proteins, thereby expediting viral replication. The lower degradation of the Cys38-Cys38 dimerized 3A protein by the proteasome [22] ultimately increases the availability of the 3A protein at the sides of viral replication, as shown here by immunofluorescence and differential centrifugation. The increase in protein 3A levels at the viral replication side would most likely augment some of the effects of the multifunctional protein 3A and its precursor 3AB—e.g., the stimulation of the viral RNA polymerase [26]. However, the disulfide-linked dimerization of protein 3A would also offer additional modes of interaction with cellular processes by not yet characterized mechanisms.

Recently, we demonstrated enhanced LC3-lipidation in cells with the ectopic overexpression of the wt 3A protein in comparison to the mutant C38A protein 3A [22], suggesting that the C38-mediated dimerization of 3A protein augments the utilization of autophagy components for viral replication. In fact, the exploitation of autophagy for viral replication is common among RNA viruses [27], and they have developed mechanisms to circumvent the eventual lysosomal degradation [28]. We propose that due to its enhanced stability, the wt CVB3 might be more capable of utilizing autophagy components than the mutant CVB3 3A[C38A], and this might support or enhance the replication process, as reported here for the wt strain. In conclusion, we suggest that Cys38-Cys38 homodimer formation by protein 3A is not required for viral RNA replication itself, but serves as a virulence factor, enlarging the sources of protein 3A at the sides of viral replication in infected cells.

## 4. Materials and Methods

### 4.1. Mice

C57BL/6J mice were obtained from a stock breeding initially purchased from Jackson Laboratory and kept at the animal facilities of the Charité—Universitätsmedizin Berlin. Five–six-week-old male mice were infected with 1 × 10^5^ plaque-forming units (pfu) of Coxsackievirus B3 strain H3 (CVB3) or the mutant CVB3-3A[C38A] by intraperitoneal injection. Three days after infection, anesthetized mice were sacrificed. Serum was obtained by the centrifugation of whole blood at 10,000 rcf for 15 min and was stored at −80 °C. Serum levels of alkaline phosphatase (AP) were determined by an external veterinary diagnostic laboratory (Vetlab, Berlin). Blood glucose levels were measured in serum samples using an AccuChek glucometer (Roche, Basel, Switzerland). Liver tissue was incubated in HistoFix (1 × PBS, 4% Roth^TM^Histofix, Carl Roth, Karlsruhe, Germany) overnight and embedded in paraffin. To visualize cellular injury and inflammation, cross sections were stained with hematoxylin and eosin. Periodic acid-Schiff histochemical analysis was performed to visualize glycogen in liver tissue sections. Tissue slides were immersed in a 0.5% periodic acid solution for 5 min, followed by incubation with Schiff’s reagent for 15 min. Counterstaining was performed with Mayer’s hematoxylin. PAS-positive cells were scored as described in [6].

### 4.2. Cell Culture

HeLa cells (ATCC) were maintained in DMEM supplemented with 10% (*v*/*v*) fetal bovine serum and 1% (*v*/*v*) penicillin/streptomycin. Phase contrast and GFP fluorescence images of cell cultures were acquired using a PAULA imager (Leica Microsystems, Wetzlar, Germany). For transient transfections, cells were grown to 90% confluence and transfected with 0.5 µg expression vector per 1 × 10^5^ cells using Polyethylenimine-Linear, MW 25,000 (Polysciences, Inc., Warrington, PA, USA). Medium was replaced 8 h after transfection.

### 4.3. Generation and Quantification of Infectious Viral Particles

The virus strains CVB3(wt) (used for all in vivo experiments) and GFP-CVB3 (used for all cell culture experiments) were generated by the transfection of HEK293T cells with the virus-cDNA containing plasmids pBKCMV-H3 (kindly provided by Andreas Henke, Friedrich Schiller University, Jena, Germany) and pMKS1-eGFP-CVB3 (provided by Zhao-Hua Zhong, Harbin Medical University, Harbin, China). HEK293T cells were transfected using PEI (polyethylenimine) Max (Polyciences, Warrington, PA, USA) and subsequently to virus-induced cell lysis, virus was amplified in HeLa cells. CVB3(wt)-3A[C38A] and CVB3-3A[C38A] were created by the site-directed mutagenesis of pBKCMV-H3 and pMKS1-eGFP-CVB3 using the primer CVB3-3A[C38A]-sense 5′-c cgt gag aga gta tgc caa aga aaa ggg atg g-3 and CVB3-3A[C38A]-antisense 5′-c cat ccc ttt tct ttg gca tac tct ctc acg g-3′. GFP-CVB3-3A[C38S] was created by the site-directed mutagenesis of pMKS1-eGFP-CVB3 using the primer CVB3-3A[C38S]-sense 5′-c cgt gag aga gta ttc caa aga aaa ggg atg g-3 and CVB3-3A[C38S]-antisense 5′-c cat ccc ttt tct ttg gaa tac tct ctc acg g-3′. PCR was run using the Q5^®^ High-Fidelity DNA Polymerase (New England Biolabs, Frankfurt am Main, Germany). Virus titers were determined by plaque assay and aliquots stored at −80 °C. Plaque assays were performed on sub-confluent monolayers of HeLa cells incubated with serial 10-fold dilutions of cell culture supernatant. After incubation at 37 °C for 30 min, supernatants were removed and monolayers were overlaid with agar containing Eagle’s minimal essential medium (MEM) and 10% FCS. After 2 days, virus plaques were stained with 0.5% MTT/PBS (3-(4,5-dimethylthiazol-2-yl)-2,5-diphenyltetrazolium bromide; Sigma).

### 4.4. Cell Lysis

Adherent cultures of HeLa cells were scraped off, pelleted by centrifugation at 2000× *g* rcf for 3 min, washed in PBS, and pelleted again. Then. the cell pellet was lysed in 20 mM Hepes pH 7.4, 1% (*v*/*v*) Triton X-114, 8 mM EDTA, 2 mM EGTA, complete protease inhibitor (Roche, Basel, Switzerland), 50 mM NaF, 5 mM Na-pyrophosphate, 2 mM Na-o-vanadate, and 10 mM NEM. After incubation on ice for 20 min, lysates were centrifuged at 16,000× *g* rcf (4 °C) for 10 min to pellet debris. Protein concentration was determined by the Bradford assay.

### 4.5. Differential Centrifugation

Adherent cultures of HeLa cells were scraped off, pelleted by centrifugation at 200 rcf for 3 min, washed in PBS, and pelleted again. Then, they were resuspended in 20 mM HEPES pH 7.4, 10 mM KCl, 2 mM MgCl2, 1 mM EDTA, 1 mM EGTA, complete protease inhibitor (Roche, Basel, Switzerland), and 10 mM NEM. Cell suspension was incubated on ice for 10 min and then passed 10 times through a 30-gauge needle. Intact cells were pelleted by centrifugation at 200 rcf for 5 min (4 °C) and the resulting supernatant transferred to a new tube for Bradford assay and subsequent differential centrifugation. Crude lysate with 2 µg/µL protein was centrifuged at 16,000× *g* rcf for 10 min (4 °C), then the supernatant was transferred to polycarbonate tubes and centrifuged at 120,000× *g* rcf (Beckman Coulter Optima TLX) for 60 min (4 °C). The supernatant and membrane pellets were immediately used for further analysis.

### 4.6. Immunoblotting

SDS-PAGE was performed on 12%, 15%, or 4–15% (Bio-Rad Laboratories GmbH, Feldkirchen, Germany) Tris-glycine gels using Tris-glycine running buffer. For SDS-PAGE, protein samples were prepared in 62.5 mM Tris HCl pH 6.8, 10% glycerol, 2% SDS, 0.005% Bromophenol Blue. The transfer of proteins onto 0.2 µm nitrocellulose membrane (Licor Bioscience, Lincoln, NE, USA) was carried out using Towbin buffer for tank blotting (Bio-Rad Laboratories GmbH, Feldkirchen, Germany) or discontinuous Tris-CAPS buffer for semi-dry blotting (Bio-Rad). Immunostaining was performed according to standard protocols. The following primary antibodies were used: GAPDH (Thermo Scientific & Abcam, Cambridge, UK), VP1 (Mediagnost, Reutlingen, Germany), and ACBD3 (Santa Cruz Biotechnology, Dallas, TX, USA). Anti-3A antibody was a gift from J. L. Whitton (The Scripps Research Institute, USA). Secondary IRD680CW or IRDye800CW labeled antibodies (Li-Cor Biosciences, Lincoln, NE, USA) were visualized using an Odyssey CLx imager and analyzed with the Image Studio software 5.2 (Li-Cor Biosciences, Lincoln, NE, USA).

### 4.7. Immunofluorescence

For immunofluorescence microscopy, cells were seeded on 13 mm cover slips coated with poly-lysine (Sigma–Aldrich, Saint Louis, MO, USA). For the staining of anti-3A (rabbit polyclonal antibody provided by K. Klingel, University of Tübingen, Germany) and anti-RCAS1 (Cell Signaling) immunofluorescence, cells were fixed with 4% paraformaldehyde in PBS for 20 min and then rinsed with PBS. After permeabilization with 0.2% Triton X-100 for 10 min, non-specific binding sites were blocked with 4% fetal bovine serum in PBS supplemented with 0.1% Triton X-100 for 30 min. Incubation with primary antibody was performed in blocking solution at room-temperature for 2 h. After three washing steps with PBS, samples were incubated with Alexa Fluor 568 coupled secondary antibody (Thermo Fisher Scientific, Waltham, Massachusetts, USA) in blocking solution for 1 h. After three washing steps with PBS and a rinse in ultra-pure water, samples were mounted on microscope slides using ROTI-Mount Fluoro-Care DAPI (Carl Roth). Images were acquired on a Nikon Scanning Confocal A1Rsi+ (Nikon, Minato, Japan) using a Plan Fluor 63× Oil objective (NA = 1.3).

For the quantitative comparison of immunofluorescence intensity distributions, confocal images in an 8-bit grey scale were analyzed using ImageJ. The area of a cell excluding the nucleus was defined as region of interest to determine the intensity distribution of anti-3A immunofluorescence. The intensities of the anti-3A staining in the region of interest were normalized to its mean intensity and intensity distributions were represented as the percentage of pixels of a particular normalized intensity.

### 4.8. Autocorrelation-Based Image Correlation Spectroscopy (ICS) of Anti-3A Immune-Fluorescence

Autocorrelation-based ICS was applied to confocal images in order to compare cluster sizes. Regions with protein 3A signal were automatically selected by Otsu’s method. Images were then auto-correlated after shifting them pixel-wise against themselves along the x and y axes. Pearson coefficients were calculated for each shift and plotted against the shift. In this analysis, large and small structures can be readily distinguished by their broad or narrow distribution of auto-correlation values, respectively. The custom written python script is available as a Jupyter Notebook: https://github.com/ngimber/ImageCorrelationSpectroscopy/releases/tag/1.0.0 (accessed on 15 March 2022).

### 4.9. Thin Section Electron Microscopy

HeLa cells were seeded into small (2 × 2.5 mm) chambers on a plastic dish (micro-insert 4 well µ-dish, ibidi, Gräfelfing, Germany) and infected with GFP-CV3B-3A wt or GFP-CV3B-3A[C38A] at an MOI of 5. After 5 h of incubation, the medium was removed and cells were fixed with 2.5% glutaraldehyde in sodium cacodylate buffer (2.6 mM MgCl2, 2.6 mM CaCl2, 50 mM KCl, 2% sucrose, pH 7.4) at room temperature. Post-fixation, en bloc contrasting, dehydration, and infiltration with epon resin (using acetone/resin mixtures) were conducted within the chambers following a standard protocol with tannic acid [29]. Thin sections (60–70 nm) were produced with an ultramicrotome (Ultracut UCT, Leica Microsystems, Wetzlar, Germany), collected on naked mesh grids, contrasted with uranyl acetate and lead citrate, and coated with a thin layer of carbon. Electron microscopy was performed with a transmission electron microscope (Tecnai Spirit, Thermo Fisher Scientific, Electron Microscopy Solutions, Eindhoven, The Netherlands) operated at 120 kV. Images were recorded with a side-mounted CCD camera (Megaview III, EMSIS, Muenster, Germany) using the image montage function to increase the pixel number.

### 4.10. Statistics

Statistical analysis of the data was performed in GraphPad Prism v7.00/v8.00 (GraphPad Software, San Diego, CA, USA). Data summary is depicted as mean ± standard error of the mean (SEM). Unpaired *t*-test was used for two group comparisons. If samples had unequal variances (determined by an F test), an unpaired *t*-test with the Welch correction was used. If values were normalized to an internal control, one-sample *t*-tests were applied. For multiple group comparison, unequal variance versions of ANOVA (one-way or two-way ANOVA) were performed followed by Sidak’s or Tukey’s multiple comparison test. The significance threshold for all tests was set at the 0.05 level.

## Figures and Tables

**Figure 1 viruses-14-00769-f001:**
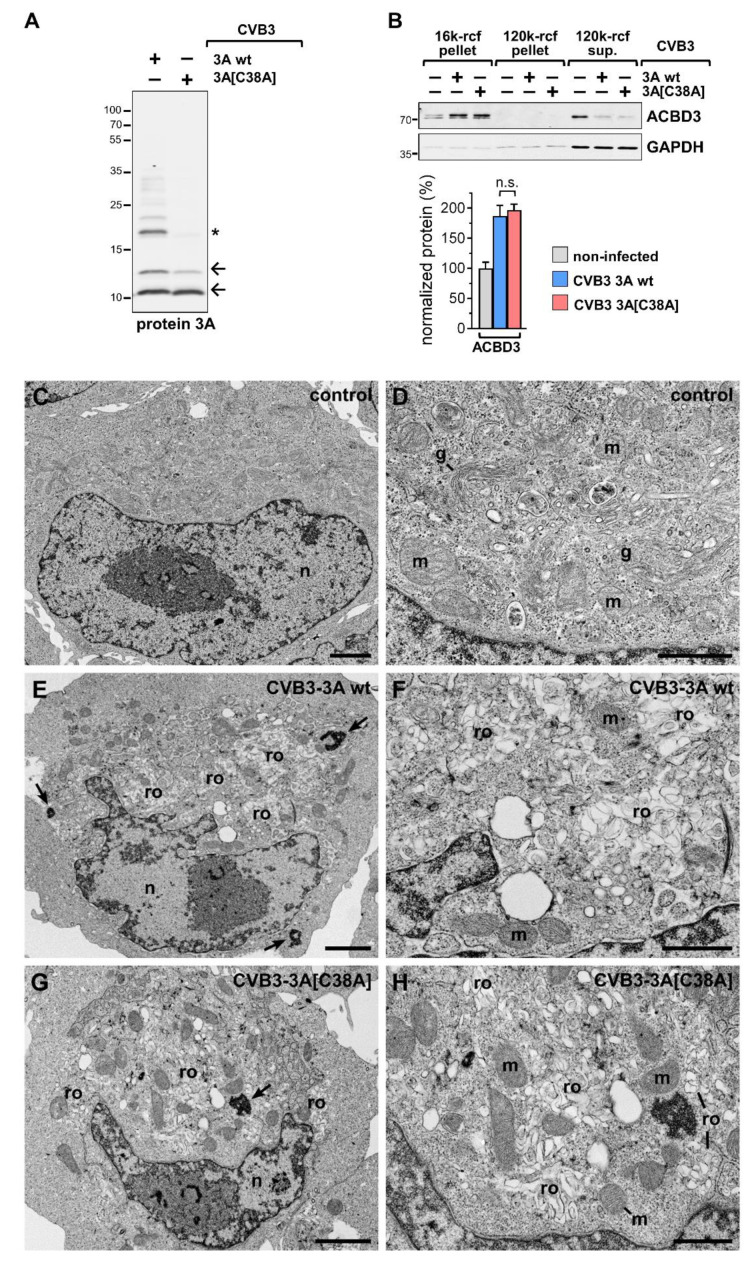
Effect of cysteine 38 of protein 3A at the ultrastructural level. (**A**) HeLa cells were infected with GFP-CVB3 encoding protein 3A wt or 3A[C38A] at MOI 1 for 16 h and processed for the immunoblotting of protein 3A. (**B**) Fractionation of non-infected cells and cells infected with GFP-CVB3 variants (3A wt and 3A C38A) at MOI 1 for 5 h. Pre-cleared lysates in hypotonic lysis buffer were subjected to differential centrifugation at 16,000 rcf (16 k) and 120,000 rcf (120 k). Resulting membrane pellets and supernatants were analyzed by the immunoblotting of 3A-interactor ACBD3, loading control GAPDH. Quantification of indicated immunosignals in 16k-rcf membrane pellets (*n* = 4). n.s.—not significant. (**C**–**G**) Thin-section electron microscopy of HeLa-cells after infection with either GFP-CVB3-3A wt or GFP-CVB3-3A[C38A]. (**C**,**D**) Non-infected cells (control). (**E**–**H**) Cells infected with GFP-CVB3-3A wt (**E**,**F**) or GFP-CVB3-3A[C38A] (**G**,**H**). Images in the right column show a detail of the overview images in the left column. Both viruses induce the formation of large virus factories, which are composed of membrane-enveloped replication organelles (ro) and dense cytoplasmic deposits (arrows) that are most likely formed by viral proteins. g: Golgi apparatus; m: mitochondrion; n: nucleus. Scale bar in (**C**,**E**,**G**) = 2 µm. Scale bar in (**D**,**F**,**H**) = 1 µm. * *p* < 0.05.

**Figure 2 viruses-14-00769-f002:**
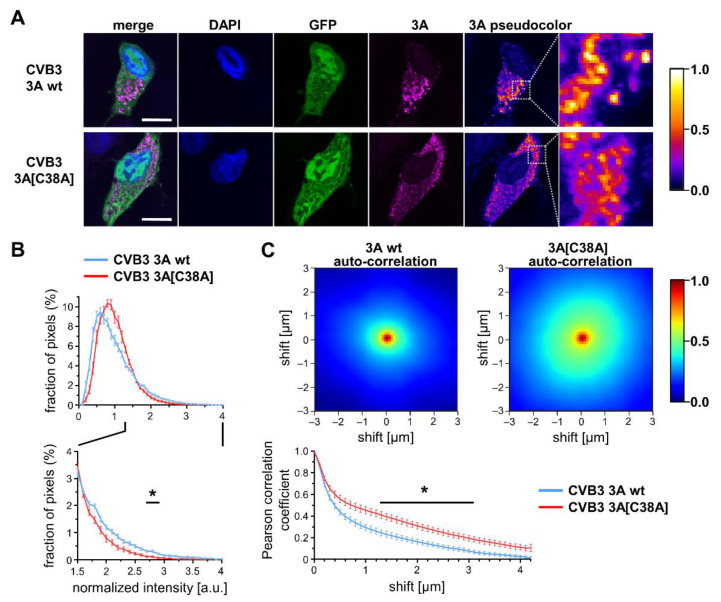
Cysteine 38 of protein 3A affects the subcellular distribution of the viral 3A protein. (**A**) Confocal microscopy of HeLa cells infected with GFP-CVB3 variants at MOI 1 for 5 h. GFP signal reflecting CVB3 replication, staining of DNA (DAPI, blue), and protein 3A (magenta). Scale bar: 10 µm. (**B**) Quantitative comparison of anti-3A immunofluorescence intensity distributions (*n* = 15) depicted in (**A**). (**C**) Autocorrelation-based image correlation spectroscopy of anti-3A immunofluorescence shown in (**A**). Two-dimensional plots depicting the auto-correlation of x/y-shifted confocal images to assess cluster sizes, which are collapsed into a 1D Pearson correlation coefficient plot (*n* = 15). In order to quantitatively compare the immunofluorescence frequency distributions and auto-correlation curve of the two strains, differences between single bins from GFP-CVB3 3Awt and GFP-CVB3 3A[C38A] were tested with a one-sample *t*-test against zero, * *p* < 0.05. We used a significance level of 0.05 and Bonferroni correction to correct for the multiple testing of several bins.

**Figure 3 viruses-14-00769-f003:**
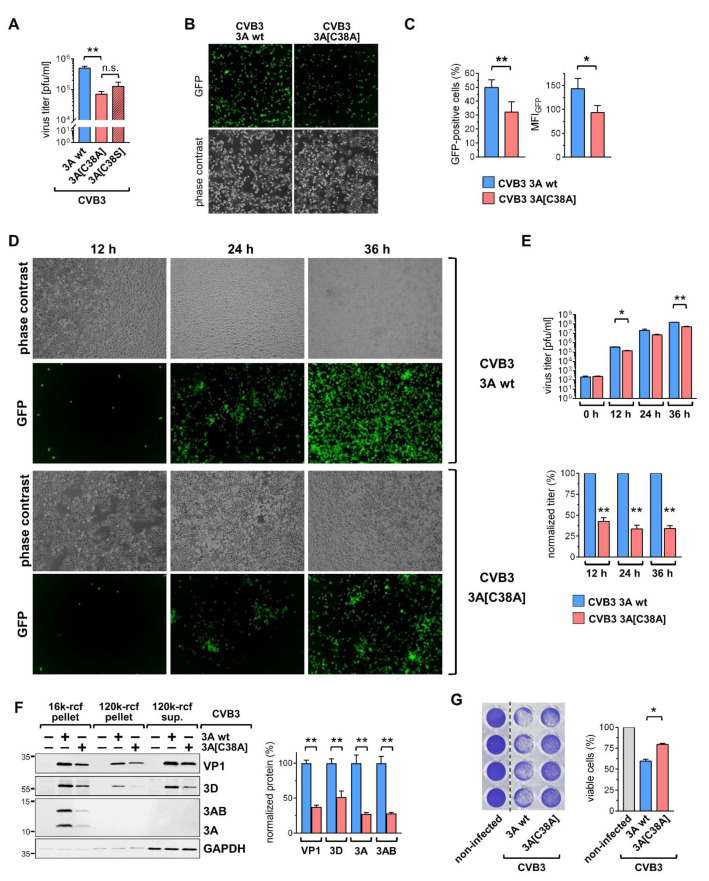
Cysteine 38 of protein 3A supports virus replication. (**A**) HeLa cells were infected with GFP-CVB3 encoding wt protein 3A, 3A[C38A], or 3A[C38S] at MOI 1 for 5 h. Virus titers were determined by plaque assay. (**B**) HeLa cells were infected with GFP-CVB3 variants at MOI 1 for 5 h. Fluorescence microscopy with the 488 nm excitation of infected cells. (**C**) Analysis by flow cytometry to assess the percentage of GFP positive cells and mean fluorescence intensity (MFI) of GFP in GFP-positive cells. (**D**) Disulfide-linked enteroviral protein 3A promotes viral replication. HeLa cells were infected with GFP-CVB3 encoding protein 3A wt or 3A[C38A] at MOI 0.01 for the indicated periods. Fluorescence microscopy at 30× magnification with 488 nm excitation of infected cells. (**E**) The amount of infectious virus was determined by plaque assay (*n* = 3). The bar charts depict the actual viral titers and normalized titers, respectively. (**F**) Fractionation of non-infected cells and cells infected with GFP-CVB3 variants at MOI 1 for 5 h. Pre-cleared lysates in hypotonic lysis buffer were subjected to differential centrifugation at 16,000 rcf (16 k) and 120,000 rcf (120 k). Resulting membrane pellets and supernatants were analyzed by the immunoblotting of CVB3 proteins 3A, 3D, and VP1. Loading control GAPDH. Quantification of indicated immunosignals in 16 k-rcf membrane pellets (*n* = 4). * *p* < 0.05. ** *p* < 0.005. n.s. non significant (**G**) Cell killing assay. HeLa cells were infected at MOI 0.5 for 24 h. Cells were fixed with 5% trichloroacetic acid and stained with crystal violet to assess cell viability (*n* = 2, each in quadruplicates).

**Figure 4 viruses-14-00769-f004:**
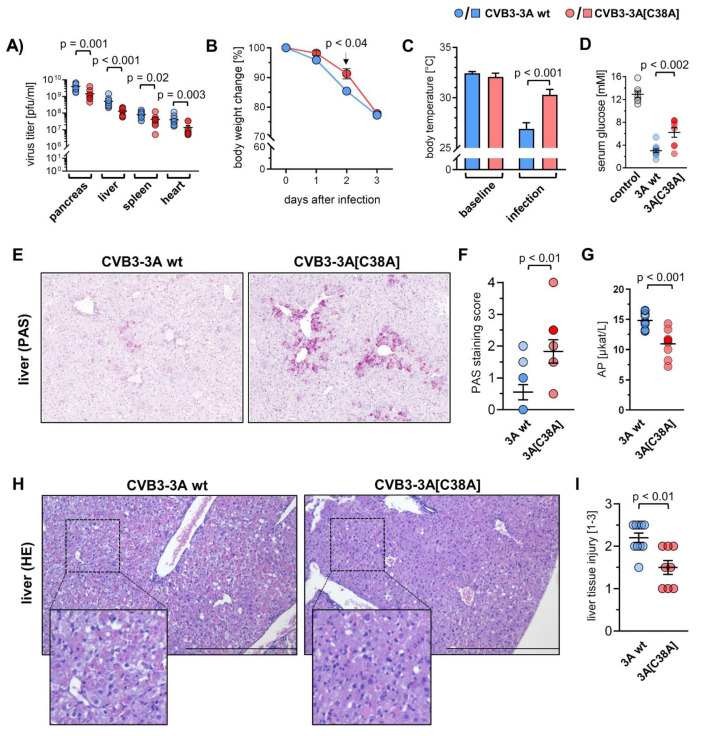
Mitigation of virus-triggered liver pathology by the CVB3-3A[C38A] mutant. Male C57BL/6 mice were infected intraperitoneally with 1 × 10^5^ pfu of CVB3 (wt) encoding wt protein 3A or 3A[C38A] and sacrificed at day 3 after infection (*n* = 10 for each virus strain). (**A**) Virus titers in indicated tissues were determined by a plaque assay. (**B**) Normalized body weight change (day 0 = 100%) at each day after infection. (**C**) Body temperature at baseline and 3 days after infection. (**D**) Serum glucose levels at baseline (control) and 3 days after infection. (**E**) Periodic acid-Schiff (PAS) staining of liver sections. (**F**) Graph depicts scoring of staining intensity (0–4, 0 = PAS-negative cells, 4 = PAS-positive cells). (**G**) Serum alkaline phosphatase levels 3 days after infection (*n* = 10 per group). (**H**) Hematoxylin-eosin staining of liver sections. (**I**) Liver pathology was categorized based on the severity of inflammation and necrosis using the following scoring system: 0: no inflammation/ necrosis; 1: scattered immune cells/mild necrosis (<10%); 2: immune cell foci/marked necrosis (10–50%); 3: diffuse immune cell infiltrates/severe necrosis (>50%), according to [23]. Unpaired *t*-test with the exception of (**B**,**C**), analysis by 2-way ANOVA with Sidak post hoc test.

## Data Availability

Not applicable.

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
