# Peer review of "A Conserved Cysteine Residue in Coxsackievirus B3 Protein 3A with Implication for Elevated Virulence"

_viruses, 2022, doi:10.3390/v14040769_

Round 1

Reviewer 1 Report

In this manuscript Voss and colleagues build on their previous work with CVB3 protein 3A  by determining the functional role of residue cysteine 38. To this end, in Figure 1 and 2, they show that though C38A mutation does not disrupt the membrane rearrangements induced by CVB3 3A as compared to wt, the mutation affects the subcellular localization of protein 3A and affects the ability of the proteins to forms more condensed structures.

In figure 3 the authors further characterize the mutation and show that a mutation at residue 38 impairs viral replication by 50% and results in a lower expression of GFP and a lower CPE. Further, in vivo experiments found that there was significantly lower virus concentration in the spleen, pancreas, heart, and liver of animals infected with the mutant as compared to CVB3-3A wt in figure 4.

Overall, this was an interesting paper and a well-organized manuscript. I have no serious concerns with the methodologies or any major issues with the manuscript in general.

However, some minor points need to be addressed:

  1. On line 209, the authors state that “Virus bearing the C38A mutation was viable and did not show ‘substantial’ growth defect at a temperature of 37°C.” However, in the next few lines they present findings that both the C38A and C38S mutants have around 50% lower viral tires that is statistically significant. Would that not be characterized as a substantial difference?
  2. In Fig 3 D, the order of the phase contrast and GFP panels is inconsistent. Please flip either one to make the order consistent.

Reviewer 2 Report

This an interesting study of viral protein 3A of coxsackievirus to promote viral replication revealed that the amino acid at position 38 on viral protein 3A of coxsackievirus binds to the same viral protein 3A to form a dimer to increase protein stability and reduce protease degradation of viral protein 3A, thus promoting coxsackievirus replication and increasing virus transmission and tissue damage in mice, This data may provide a basis for further discussion on which inhibitors or natural compounds can effectively inhibit the activity of coxsackievirus viral protein 3A to suppress virus replication and infection. However, in the light of these interesting inferences, there are some issues that need to be clarified in the text

I want to confirm that the viral protein 3A of coxsackievirus is mainly due to the fact that the amine at position 38 binds to the homologous viral protein 3A to form a dimer to increase protein stability, resulting in the inability of the protease to degrade it, so that viral protein 3A can steadily induce the binding of ACBD3 for virus replication, and not for other reasons. (The main reason is that from the text and discussion, it is clear that viral protein 3A does not affect the formation of viral replication organelles and induce the binding of ACBD3 to viral protein 3A after the mutation of amino acid at position 38).

3. In addition, I would like to ask the authors whether they will conduct the same experiments on the other two proteins (GBF1 and PI4KB) that were mentioned in the discussion as being useful for viral RNA replication and viral particle assembly to demonstrate that viral protein 3A will bind to these two proteins to promote viral replication.

Reviewer 3 Report

While this is a good experimental paper, I have some concerns.

The authors basically attribute the attenuation of their mutated EV sample to the role of  Protein 3 A  in recruiting proteins to the Golgi  towards the packaging of infectious particles. While not arguing against the validity of this hypothesis and the experiment, you will find other hypotheses in a literature search:

https://www.jbc.org/article/S0021-9258(18)92075-2/fulltext

This article is basically saying that protein AB plays a role in stimulating the RNA polymerase in the viral RNA replication.

1) While such should not be surprising as viral proteins are often multifaceted and multifunctional,  did the authors look at other hypotheses such as this?  Could the mutation be actually affecting the stimulation of the RNA polymerase? The authors may want to do further literature search to address the possibility

2)  As mentioned, viral proteins are often multifaceted and multifunctional. There could be different domains within a protein that are responsible for different modes of action. Sometimes domains with separate functions may overlap each other:

https://pubmed.ncbi.nlm.nih.gov/25105276/

This question should be addressed in the paper

3). I noticed that the loss of virulence involves a mutation from cysteine to alanin- i.e.  a polar residue to non-polar one. Polar residues tend to induce disorder and disorder allows greater fits in protein-protein/RNA/lipid interactions. This is not something new. Greater disorder in viral proteins that are intimately involved in viral replication has been shown to induce virulence in a wide variety of virus such as sars-cov-1/2, ebola, nipah and dengue viruses:

https://pubmed.ncbi.nlm.nih.gov/35142523/

Round 2

Reviewer 3 Report

Improvement seen